# Validity of a Portable Breath Analyser (AIRE) for the Assessment of Lactose Malabsorption

**DOI:** 10.3390/nu11071636

**Published:** 2019-07-17

**Authors:** Aahana Shrestha, Utpal K. Prodhan, Sarah M. Mitchell, Pankaja Sharma, Matthew P.G. Barnett, Amber M. Milan, David Cameron-Smith

**Affiliations:** 1The Liggins Institute, The University of Auckland, 85 Park Road, Grafton, Auckland 1023, New Zealand; 2The Riddet Institute, Palmerston North 4442, New Zealand; 3Department of Food Technology and Nutritional Science, Mawlana Bhashani Science and Technology University, Tangail 1902, Bangladesh; 4Food Nutrition & Health Team, AgResearch Limited, Private Bag 11008, Palmerston North 4442, New Zealand; 5The High-Value Nutrition National Science Challenge, Auckland 1023, New Zealand; 6Food & Bio-based Products Group, AgResearch Limited, Private Bag 11008, Palmerston North 4442, New Zealand

**Keywords:** carbohydrate malabsorption, breath hydrogen, personal health device, milk, dairy

## Abstract

Hydrogen (H_2_) measurement in exhaled breath is a reliable and non-invasive method to diagnose carbohydrate malabsorption. Currently, breath H_2_ measurement is typically limited to clinic-based equipment. A portable breath analyser (AIRE, FoodMarble Digestive Health Limited, Dublin, Ireland) is a personalised device marketed for the detection and self-management of food intolerances, including lactose malabsorption (LM). Currently, the validity of this device for breath H_2_ analysis is unknown. Individuals self-reporting dairy intolerance (six males and six females) undertook a lactose challenge and a further seven individuals (all females) underwent a milk challenge. Breath samples were collected prior to and at frequent intervals post-challenge for up to 5 h with analysis using both the AIRE and a calibrated breath hydrogen analyser (BreathTracker, QuinTron Instrument Company Inc., Milwaukee, WI, USA). A significant positive correlation (*p* < 0.001, r > 0.8) was demonstrated between AIRE and BreathTracker H_2_ values, after both lactose and milk challenges, although 26% of the AIRE readings demonstrated the maximum score of 10.0 AU. Based on our data, the cut-off value for LM diagnosis (25 ppm H_2_) using AIRE is 3.0 AU and it is effective for the identification of a response to lactose-containing foods in individuals experiencing LM, although its upper limit is only 81 ppm.

## 1. Introduction

Food intolerance is an adverse reaction to food without direct involvement of the immune system, which affects 15%–20% of the population [1]. Incomplete digestion of short-chain carbohydrates and polyols (fermentable oligosaccharides, disaccharides, monosaccharides and polyols, or FODMAPs) results in anaerobic fermentation in the intestine, liberating hydrogen (H_2_) which subsequently diffuses into the bloodstream prior to release in the breath [2]. One of the common examples of carbohydrate malabsorption occurs in response to lactose, a disaccharide in dairy products. Worldwide, 75% of the adult population have limited expression of the small intestinal brush border enzyme β-lactase [3] either due to genetically determined lactase non-persistence [4] or secondary to other gastrointestinal disorders; in both cases, this leads to lactose malabsorption (LM) [5]. A proportion of these people will experience adverse digestive symptoms due to the lactose fermentation [6]. These adverse symptoms include abdominal discomfort, bloating, pain, faecal urgency, and diarrhoea [7,8], which can have negative impacts on quality of life, and prompt self-directed changes in dietary behaviour and intake [9].

Breath H_2_ is the most widely used non-invasive method to diagnose carbohydrate malabsorption [10,11] including LM. Diagnosis of LM is typically dependent upon the measurement of breath H_2_ using a breath gas analyser, where an increase in breath H_2_ from baseline of more than 25 ppm after lactose ingestion (50 g) at any postprandial time is considered diagnostic for LM [12]. Although there are different methods and instruments available, these are typically specialised and expensive, requiring both technical support and regular calibration using standard gas mixtures to ensure accuracy [13]. Hence, lactose and associated meal-challenge tests to establish LM are typically performed in diagnostic clinical settings. This precludes regular and personalised identification of possible LM or lactose-containing foods during daily living. 

Personalised health technologies are increasingly enabling regular self-monitoring of health-related signs [14]. The expanding range of personalised devices is marketed to help in achieving health benefits and sustaining behavioural changes. Current technologies are widely aimed at improving fitness gains and for improved self-management of hypertension [15,16] and type 2 diabetes [17]. In 2018, a portable pocket-sized breath analyser, the AIRE, was introduced into this innovative and expanding market by FoodMarble Digestible Heath Limited (Dublin, Ireland) [18]. The AIRE device is claimed to be the world’s first personal real-time digestive tracker [19]. It is reported to measure exhaled breath H_2_ and, together with a smartphone, enables the user to wirelessly (via Bluetooth) transfer data from the device to a personalised application (App). This, when combined with the App’s diary function, provides the individual with the opportunity to monitor food intake, timing, AIRE values, and symptom onset. The AIRE values are provided as a fermentation score (scale 0–10), reported by the company to be based on proprietary technologies of H_2_ detection [19]_._

To date, there are no reported studies describing the accuracy or application of the AIRE device for assessing breath H_2_ with LM. Therefore, the aim of this study was to analyse the relationship between data from the AIRE and data obtained using an established breath H_2_ analyser. The analysis was conducted in adults with suspected LM following the consumption of lactose or milk, to provide relevance to both clinical and home applications. 

## 2. Materials and Methods

### 2.1. Study Design

Nineteen healthy individuals aged 20–40 years with a body mass index (BMI) of 18–28 kg/m^2^ were recruited from 2 different cohorts having similar inclusion and exclusion criteria but using a different substrate to assess LM. The primary outcomes of these studies are reported elsewhere. Individuals consuming antibiotics 3 months prior to the study, having inflammatory bowel disease, or with a known milk allergy were excluded from the study. No subjects reported cardiovascular or metabolic disease. One cohort included self-reported dairy intolerant individuals (n = 12, 6 males and 6 females) and used lactose to assess LM. The second cohort included only females avoiding dairy (n = 7) and used milk as substrate to measure LM. The study was conducted according to the guidelines laid down in the Declaration of Helsinki and all procedures involving human subjects were approved by the New Zealand Health and Disability Ethics Committees (applications 16/STH/175 and 18/NTB/92). All participants provided written informed consent before being enrolled in the studies. Both cohorts were from studies registered with the Australian New Zealand Clinical Trials Registry (IDs: ACTRN12616001694404 and ACTRN12618001030268).

### 2.2. Study Procedure

All subjects attended the Maurice and Phyllis Paykel Clinical Research Unit at the Liggins Institute, The University of Auckland between October and December 2018. One day prior to the clinical visit, subjects were advised to abstain from vigorous physical exercise, avoid dairy or fibre rich food, and were provided with a standardised low fat, low dietary fibre evening meal. Subjects were instructed to remain fasted from 10.00 p.m. the night prior to the visit.

Upon arrival, fasting breath samples were collected twice for analyses, once using the AIRE and then within 2 min using the BreathTracker H2+ model (QuinTron Instrument Company Inc, Milwaukee, WI, USA). Subjects were challenged with either 50 g lactose (dissolved in 250 mL water) or 650 mL of either sheep or cow milk (prepared from full cream milk powder containing approximately 32 g of lactose). Breath samples were collected postprandially every 15 min until 90 min then hourly until 4 h after milk, and every 30 min until 2 h and hourly until 5 h after lactose ingestion and analysed using both devices.

### 2.3. Breath Analysis

All breath sampling was performed in a standardised manner using the instructions provided by each manufacturer. For the AIRE readings, subjects partially inhaled, then held their breath for 3 s before slowly exhaling into the mouthpiece for a period of 5 s. H_2_ concentration was represented as a fermentation score which ranged from 0.0–10.0 arbitrary units (AU; based on a propriety ppm algorithm) in the connected AIRE App. The AIRE device also provided some qualitative data including ‘low’, ‘okay’, or ‘high’ for each H_2_ value, but these data were not collected for the current study. For the BreathTracker, expired breath samples were collected using the AlveoSampler^TM^ (QuinTron) breath test kit in a 30 mL plastic syringe and then analysed in the BreathTracker. Data were collected as CO_2_-corrected H_2_ expressed in parts per million (ppm). The maximum increase in breath H_2_ (max delta) was calculated by subtracting baseline H_2_ values from the highest H_2_ value post lactose ingestion based on BreathTracker readings.

### 2.4. Statistical Analysis

The sample size for similar studies reporting breath H_2_ have used a higher sample size but have not reported the correlation coefficient (r) [20,21]. So sample size was calculated based on published results from correlation of home-based and laboratory-based methods [22]. To provide an 80% power with alpha set at 5%, based on previously reported correlation (r = 0.9), 7 subjects would be required. Correlations between breath H_2_ concentrations from both AIRE and BreathTracker devices were calculated using Pearson’s correlation and simple linear regression analysis after lactose and milk ingestion separately. Due to the differing units between the two devices a Bland–Altman plot was not computed. All statistical analyses were performed using Graph Pad Prism version 7.03 (GraphPad Software, La Jolla, CA, USA). A *p* value less than 0.05 was considered statistically significant. Mean and standard deviation (SD) of breath H_2_ from both the devices were calculated pre (fasting) and post lactose and milk ingestion. Max delta breath H_2_ was calculated by subtracting the maximum H_2_ obtained after lactose challenge from the baseline H_2_ values. The sample size was calculated based on correlation of the two methods.

## 3. Results

In total 136 breath H_2_ measurements were made using both devices; 96 to assess the response after the lactose challenge and 40 to assess the response after the milk challenge. This included 21 readings at baseline (Table 1).

Following an overnight fast, the average reading (mean ± SD) for the AIRE was 0.6 ± 0.4 AU and for the BreathTracker 8 ± 5 ppm. Post lactose and milk ingestion, there was an increase in breath H_2_ recorded using both devices (Figure 1). The average reading for the AIRE after lactose ingestion was 6.0 ± 3.5 AU and after milk ingestion was 5.6 ± 3.9 AU. Similarly, the corresponding Quintron reading was 66 ± 59 and 52 ± 46 ppm after lactose and milk ingestion respectively. As a peak H_2_ value was only obtained by the Quintron in seven and four participants for the lactose and milk challenges respectively, the max delta was calculated to show the relative increase from baseline readings. This max delta (Mean ± SD) was 120 ± 72 ppm and 84 ± 60 ppm for the lactose and milk challenges, respectively.

The AIRE showed a maximum score of 10 AU after both lactose and milk ingestion, which is the highest possible score using this device. The maximum H_2_ reading from BreathTracker was 266 ppm after lactose ingestion and 147 ppm after milk ingestion. The results demonstrated a significant (*p* < 0.001) and positive linear correlation between the H_2_ concentration measured using the AIRE (AU) and BreathTracker (ppm) after both lactose (r = 0.82 and *p* < 0.001; Figure 1A and Table 1) and milk ingestion (r = 0.90 and *p* < 0.001; Figure 1C and Table 1). For 26% (36 of 136) of the AIRE readings, the maximum score of 10 AU was recorded. The correlation remained significant (*p* < 0.001) even following removal of the maximum AIRE score of 10 after both lactose (r = 0.77; Figure 1B) and milk ingestion (r = 0.89; Figure 1D and Table 1).

Using the equation of the line for the correlation without maximum AIRE scores (Figure 1C,D), an estimated maximum accurate detection limit was calculated. Using the lactose correlation, the maximum accurate detection limit for AIRE was equivalent to 81 ppm on the BreathTracker; for the milk correlation, this value was 65 ppm. Overall, following lactose ingestion 23 of the 28 BreathTracker H_2_ readings above 81 ppm (i.e., 82%) generated an AIRE score of 10 AU. Likewise, following the milk ingestion, 10 of the 10 BreathTracker H_2_ readings above 65 ppm (100%) had a maximum score of 10 AU from AIRE. Further, 9 of 12 subjects after the lactose challenge and 5 of 7 subjects after the milk challenge showed AIRE readings of 10 AU on at least one postprandial time point.

When only baseline values were considered, no correlation between H_2_ readings from the AIRE and BreathTracker (r = 0.08; Figure 1E) was observed. However, the maximum delta readings from the AIRE and BreathTracker were positively correlated (r = 0.77). Out of the 12 participants enrolled for the lactose challenge, we had baseline AIRE readings for only 10 participants, and 9 out of the 10 participants were diagnosed as lactose malabsorbers based on the BreathTracker reading (max delta H_2_ > 25 ppm) (Figure 1F). Using the equation of the line after the lactose challenge (Figure 1C) the cut off value of 25 ppm from BreathTracker was equivalent to an AIRE value of approximately 3.0 AU. Using 3.0 AU from AIRE as the cut-off, all 10 participants were diagnosed as lactose malabsorbers.

## 4. Discussion

Food allergies and intolerances are commonly self-diagnosed and are a widely reported health concern [17,18]. The AIRE is a portable and personalised breath H_2_ analyser that is marketed to improve self-management of digestive issues, including LM [19]. It provides information in terms of a fermentation score between 0.0 to 10 AU and hence provides a simple score of possible malabsorption. In this study, AIRE values were compared with breath H_2_ concentration (ppm) measured using a well-established and calibrated breath H_2_ analyser, the BreathTracker. For this study, individuals expected to generate breath H_2_ were recruited, thus following either lactose or milk ingestion breath H_2_ concentrations were increased for most participants. There was a highly significant correlation between the portable AIRE and the BreathTracker, however, no correlation was present at the lower threshold i.e., when only readings at baseline were considered. Furthermore, in 26% of readings using the AIRE device the score reached the maximum of 10 AU. Thus, the AIRE was not able to reliably identify the maximal extent of malabsorption in these cases. Nevertheless, the correlation between the two devices remained significant after removal of the 26% of readings showing an AIRE score of 10 AU.

The portable AIRE device is not marketed as a replacement to existing clinical breath analysers, but as an aid for individuals to track digestion in real time [18], with the aim of identifying and limiingfoods that result in maldigestion. Given the potential application to differing circumstances, the current study examined the responsiveness to both pure lactose and milk (containing lactose) [12,23]. Prior to lactose and milk ingestion, breath H_2_ reported by both the devices was very low, whereas there was a clear increase in breath H_2_ after either lactose or milk ingestion. The AIRE reported a maximum score of 10 AU whereas BreathTracker reported an H_2_ concentration in ppm, which was higher after the lactose challenge compared to the milk challenge. The higher H_2_ concentration after lactose challenge could be due to a higher dose of lactose [24]. When only baseline values were considered, no correlation existed between the measurement from AIRE and BreathTracker indicating variability at the lower range of H_2_ measurements. Nevertheless, all the baseline values were below 25 ppm which is equivalent to 3.0 AU based on the correlation equation after the lactose challenge. This indicates normal H_2_ values despite the discrepancies in the H_2_ readings from the two devices. Furthermore, the maximum H_2_ recorded from AIRE was 10 AU, with the equivalent concentration in ppm after lactose challenge being 81 ppm; after the milk challenge this corresponded to 65 ppm. In spite of this discrepancy, both values are high enough to diagnose an individual as a lactose malabsorber. Therefore, the AIRE can provide a preliminary diagnosis of LM, although its upper limit is approximately 81 ppm and the accuracy within the lower part of its range needs to be validated in a larger cohort. It should also be noted that this is in the context of appropriate preparation conditions for an H_2_ breath test, requiring adherence to food restrictions prior to using the AIRE to avoid false positives, as would be required with any other H_2_ testing device. Users may need to follow these protocols to attain reliable results, including considerations such as avoiding antibiotics for at least three months, avoiding high fibre food or dairy one day prior to the test, and having an appropriately low fasting measurement.

The correlations between readings from AIRE and BreathTracker were significant regardless of the substrate used, with a slightly higher correlation after the milk challenge compared to the lactose challenge. This supports the reliability of the AIRE when using milk as a substrate, which is a more realistic and pragmatic approach for consumers to diagnose LM rather than using pure lactose which is common practice in clinical settings. One point to note is that the BreathTracker H2+ model uses simultaneous CO_2_ measurement, which provides sample quality assurance [25]. It is possible that the AIRE lacks this correction mechanism, impacting the reliability of readings. Regardless, the correlation discrepancy between milk and lactose highlights a difference in the detection of breath H_2_ from differing substrates, which may have implications for expected readings from various mixed foods. However, these data provide preliminary evidence that the AIRE is suitable to assess LM using milk as a substrate. In addition to lactose, it may also help detect malabsorption to other FODMAPs, but the influence of different food matrices on expected readings requires further exploration.

The AIRE is a small handheld device, easy to operate, and linked to a downloadable App which enables the user to record features related to symptoms of gastrointestinal comfort and additional lifestyle information including sleep patterns [19]. The ability to monitor objective measures (breath H_2_) of malabsorption that can be correlated with any feelings of discomfort allows consumers to distinguish malabsorption-related discomfort from other symptoms, and to identify the foods causing digestive problems. In this way, consumers may be capable of using the AIRE not only for diagnosis, but also for monitoring the cause of discomfort if it returns. A similar strategy to monitor exposure to gluten has been developed, with the home test kit “Gluten Detective” launched by Glutenostics to improve the diagnosis and management of celiac disease and related digestive discomfort [26].

The AIRE provides an easy at-home preliminary diagnosis of LM, yet it is not clear how the data provided by the device will be interpreted by individuals and incorporated into their personal assessment of digestive health. The information from the device may result in individuals avoiding certain foods unnecessarily causing nutrient insufficiency. It is therefore important to access the abdominal symptoms associated with malabsorption because malabsorption is not always accompanied by such symptoms [6,9]. The malabsorption readings by themselves may lead to behaviour changes, even though malabsorption itself is a proxy of potentially beneficial bacterial fermentation in the larger context of health [27,28,29]. Some studies have shown that the avoidance of FODMAPs reduces the relative abundance of beneficial bacteria such as bifidobacteria [29,30] and increases the abundance of harmful bacteria such as *Porphyromonadaceae* [29]. However, other studies suggest low FODMAP diets aid in the reduction of symptoms in individuals with irritable bowel syndrome (IBS) and inflammatory bowel disease (IBD) [31,32]. Nevertheless, a low FODMAP diet is not sustainable, and if strictly followed it may lead to inadequate nutrient intake [33,34]. Therefore, it is important to accurately identify which FODMAPs or other fermented foods trigger malabsorption which is measurable using AIRE and to understand how this malabsorption may be used to appropriately modify dietary behaviours. Furthermore, it is important to share the results with health care professionals, although it is still necessary to determine whether or not the data provided by the device aresufficient to influence health care providers’ therapeutic strategies, or whether traditional lactose (or other carbohydrate) malabsorption testing will still be required.

## 5. Conclusions

The present study demonstrated that there is a significant positive correlation between breath H_2_ determined using a new portable breath analysis device, the AIRE, and a standard gas calibrated breath hydrogen analyser, the BreathTracker. Based on these data, the AIRE may be applicable for the assessment of breath H_2_ up to values of approximately 81 ppm, depending on the substrate. Above this concentration of breath H_2_, the AIRE is not able to provide quantitative data on breath H_2_ levels. 

## Figures and Tables

**Figure 1 nutrients-11-01636-f001:**
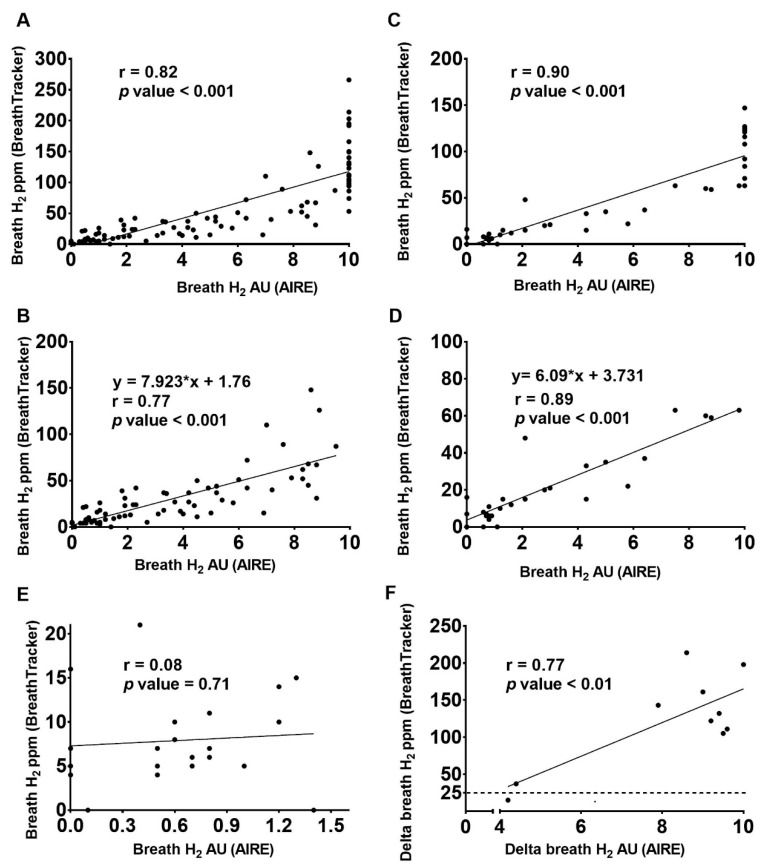
Scatterplots with simple linear regression line showing the relationship between H_2_ readings from the AIRE (x-axis) and BreathTracker (y-axis), after lactose ingestion with all readings (**A**); after lactose ingestion with AIRE scores of 10 removed (**B**); after milk ingestion with all readings (**C**); after milk with AIRE scores of 10 removed (**D**); at baseline before lactose or milk ingestion (**E**); and the max delta after lactose ingestion (**F**). Figure 1A–D represent raw H_2_ readings i.e., the baseline is not subtracted.

**Table 1 nutrients-11-01636-t001:** Linear regression analyses for the association of breath H_2_ measurement using AIRE and BreathTracker ^1^.

	No. of H_2_ Measurements	R^2^	Slope (Mean ± SD)	Pearson Correlation (r)	*p* Value
**Lactose challenge ^2^ (n = 12)**					
All readings including AIRE score = 10	96	0.67	12.64 ± 0.90	0.82	<0.001
Without AIRE score = 10	71	0.60	7.92 ± 0.77	0.77	<0.001
**Milk challenge ^3^ (n = 7)**					
All readings including AIRE score = 10	40	0.81	9.79 ± 0.76	0.90	<0.001
Without AIRE score = 10	29	0.80	6.09 ± 0.57	0.89	<0.001

^1^ Simple linear regression and Pearson correlation was computed, R^2^ represents the coefficient of determination. ^2^ Lactose challenge was performed using 50 g lactose in 250 mL of water. ^3^ Milk challenge was performed using 650 mL of milk.

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
