# Peer review of "Validity of a Portable Breath Analyser (AIRE) for the Assessment of Lactose Malabsorption"

_nutrients, 2019, doi:10.3390/nu11071636_

Round 1

Reviewer 1 Report

Comments for Authors

     This paper tests a new, portable breath hydrogen detector by comparison with the best of the standard clinical devices. The very high correlation coefficient between closely paired measurements indicates that the portable detector does an excellent job. Its simplicity of use suggests promise as a home testing device. Along with its cloud-based data support, the AIRE could be very useful in home-based research studies. Here, as the authors note, a wealth of self-reported data could allow correlation of breath hydrogen with the onset of specific symptoms in relation to meals and uncover other malabsorptions in patients.

     The authors have chosen lactose malabsorption, for which diagnosis is especially well matched to hydrogen-detection technology. The excellent performance of the AIRE system with lactose-intolerant patients promises further use with fructose intolerance and other more complex conditions leading to nutrient malabsorption by the small intestine. The paper is very well written. However, there are a few issues that should be addressed.

1.       Introduction: should include a brief review of accepted thresholds of lactose-stimulated peak breath hydrogen for a clinical diagnosis of lactose intolerance using the QuinTron Breath Tracker (or equivalents) and a lactose challenge.

2.       Methods:  Needs to note whether the AIRE also reads fractional AU units. This seems to be the case from the plots, but whether it is to the nearest 0.1 or 0.2 AU unit is not clear.  

3.       Methods:  This section needs to more clearly specify that the Breath H2 ppm in Figure 1 represents raw H2 measures, for which the baseline value for the patient has not been subtracted.

4.       Results:  The authors need to indicate, for the QuinTron data, how often baseline-corrected peak values of H2 were obtained during the measurement period. Detection of a peak value is defined as a maximum value followed by at least one data point of declining H2.  

5.       Results: One additional graph of the 21 baseline measurements should be added to Figure 1 (Figure 1. E). This will also present the actual baseline data (which is not marked in A-D). Axes should be expanded to better present how well the AIRE performs in the low end of its range.

6.       Results: A peak value graph (Figure 1. F) should be added for all lactose-challenge patients. These should similarly plot baseline-corrected paired AIRE and QuinTron values, one point for each patient. Measurements for plotting should be selected based on only peak values in the QuinTron data, since the AIRE is not as useful for measuring peak values. Figure 1E could use a horizontal dashed line to indicate the QuinTron peak value diagnostic threshold for lactose malabsorption.

7.       Results: From the peak value results, it should be possible to indicate which (if all) of these self-identified patients have a diagnosis of lactose-intolerance by the QuinTron results, and by the AIRE data.  

8.       Discussion:  This section should address how well the AIRE might be expected to provide a preliminary diagnosis of lactose intolerance.

9.       Discussion: There should be some mention of the genetics of lactose intolerance. Specifically how previous papers have correlated the inheritance of the key lactase gene polymorphism with breath test diagnosis.

Author Response

Please refer to the attachment: "nutrients-536473_Reponse-to-reviewer1_FINAL.docx"

Reviewer 2 Report

1- it is not clear what is the challenge: dosage of lactose and the milk and type of milk.

2- it is not clear why the 80ppm limit; often in practice there is far higher H2 produced during a malbsorption testing. Although it may not be necessary to have higher than this limit generally considering that only a rise of 20ppm against baseline is required to make diagnosis.

3- What was the patient preparation as it is very important to consider many factors including the dieting prior to the tests.

4- If this device is meant to be used by public, is that considered that the person has to follow the many preparation requirements ie avoiding antibiotics for 4 weeks, not having endoscopies for around 2 weeks, dieting from fermentable foods, fasting prior to the actual testing time, not physically moving or sleeping during the test etc...... ?

5- what is the mean and standard deviation of the measurements? Also, what parameter is used for comparisons: mean, median, etc? Statistical method is not clear.

6- sample size seems to be too small. has a sample size calculation done? it does not seem to be justifiable to mix and match different test methods to add up ie milk method and the substrate method.

- provided all the info for the patients to acquire an accurate result, this may be useful for public use although this is not a test the patients need to do in regular basis ie like blood pressure or blood glucose check. Why somebody need to do the testing more than once every few years?

Author Response

Please refer to the attachment: "nutrients-536473_Response-to-reviewer2_FINAL.docx"
